# Procedural generalization by planning with self-supervised world models

**Ankesh Anand**[†][*]**, Jacob Walker**[†]**, Yazhe Li**[‡]**, Eszter Vértes**[‡]**, Julian Schrittwieser, Sherjil Ozair, Théophane Weber, Jessica B. Hamrick**
DeepMind, London, UK

## Abstract

One of the key promises of model-based reinforcement learning is the ability to generalize using an internal model of the world to make predictions in novel environments and tasks. However, the generalization ability of model-based agents is not well understood because existing work has focused on model-free agents when benchmarking generalization. Here, we explicitly measure the generalization ability of model-based agents in comparison to their model-free counterparts. We focus our analysis on MuZero [60], a powerful model-based agent, and evaluate its performance on both procedural and task generalization. We identify three factors of procedural generalization—planning, self-supervised representation learning, and procedural data diversity—and show that by combining these techniques, we achieve state-of-the art generalization performance and data efficiency on Procgen [9]. However, we find that these factors do not always provide the same benefits for the task generalization benchmarks in Meta-World [74], indicating that transfer remains a challenge and may require different approaches than procedural generalization. Overall, we suggest that building generalizable agents requires moving beyond the single-task, model-free paradigm and towards self-supervised model-based agents that are trained in rich, procedural, multi-task environments.

## 1 Introduction

The ability to generalize to previously unseen situations or tasks using an internal model of the world is a hallmark capability of human general intelligence [12, 41] and is thought by many to be of central importance in machine intelligence as well [14, 26, 59, 66]. Although significant strides have been made in model-based systems in recent years [28], the most popular model-based benchmarks consist of identical training and testing environments [e.g. 27, 69] and do not measure or optimize for for generalization at all. While plenty of other work in model-based RL does measure generalization [e.g. 18, 48, 70, 76], each approach is typically evaluated on a bespoke task, making it difficult to ascertain the state of generalization in model-based RL more broadly.

Model-free RL, like model-based RL, has also suffered from both the "train=test" paradigm and a lack of standardization around how to measure generalization. In response, recent papers have discussed what generalization in RL means and how to measure it [7, 8, 36, 49, 71], and others have proposed new environments such as Procgen [9] and Meta-World [74] as benchmarks focusing on measuring generalization. While popular in the model-free community [e.g. 47, 73, 78], these benchmarks have not yet been widely adopted in the model-based setting. It is therefore unclear whether model-based methods outperform model-free approaches when it comes to generalization, how well model-based methods perform on standardized benchmarks, and whether popular model-free algorithmic improvements such as self-supervision [15, 44] or procedural data diversity [67, 77] yield the same benefits for generalization in model-based agents.

In this paper, we investigate three factors of generalization in model-based RL: planning, self-supervised representation learning, and procedural data diversity. We analyze these methods through

---

[*]Work done while visiting from Mila, University of Montreal.
[†]Joint first authors.
[‡]Equal contribution.

a variety of modifications and ablations to MuZero Reanalyse [60, 61], a state-of-the-art model-based algorithm. To assess generalization performance, we test our variations of MuZero on two types of generalization (See Figure 1): *procedural* and *task*. Procedural generalization involves evaluating agents on unseen configurations of an environment (e.g., changes in observation rendering, map or terrain changes, or new goal locations) while keeping the reward function largely the same. Task generalization, in contrast, involves evaluating agents' adaptability to unseen reward functions ("tasks") within the same environment. We focus on two benchmarks designed for both types of generalization, Procgen [9] and Meta-World [74].

Our results broadly indicate that self-supervised, model-based agents hold promise in making progress towards better generalization. We find that (1) MuZero achieves state-of-the-art performance on Procgen and the procedural and multi-task Meta-World benchmarks (ML-1 and ML-45 train), outperforming a controlled model-free baseline; (2) MuZero's performance and data efficiency can be improved with the incorporation of self-supervised representation learning; and (3) that with self-supervision, less data diversity is required to achieve good performance. However, (4) these ideas help less for task generalization on the ML-45 test set from Meta-World, suggesting that different forms of generalization may require different approaches.

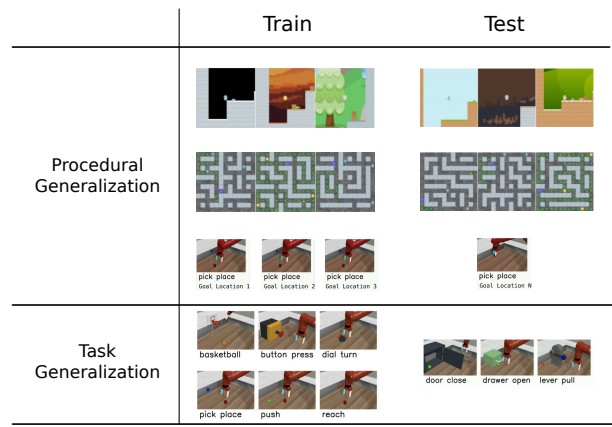

Figure 1: Two different kinds of generalization, using Procgen and Meta-World as examples. *Procedural* generalization involves evaluating on unseen environment configurations, whereas *task* generalization evaluates adaptability to unseen tasks (reward functions).

## 2 MOTIVATION AND BACKGROUND

### 2.1 GENERALIZATION IN RL

We are interested in the setting where an agent is trained on a set of $n_{\text{train}}$ MDPs drawn IID from the same distribution, $\mathbb{M}_{\text{train}} = \{\mathcal{M}_i\}_{i=1}^{n_{\text{train}}}$, where $\mathcal{M}_i \sim p(\mathcal{M})$. The agent is then tested on another set of $n_{\text{test}}$ MDPs drawn IID[1] from $p$, but disjoint from the training set: $\mathbb{M}_{\text{test}} = \{\mathcal{M}_j\}_{j=1}^{n_{\text{test}}}$ where $\mathcal{M}_j \sim p(\mathcal{M})$ such that $\mathcal{M}_j \notin \mathbb{M}_{\text{train}} \forall \mathcal{M}_j$. *Generalization*, then, is how well the agent performs in expectation on the test MDPs after training (analogous to how well a supervised model performs on the test set). Qualitatively, generalization difficulty can be roughly seen as a function of the number ($n_{\text{train}}$) of distinct MDPs seen during training (what we will refer to as *data diversity*), as well as the breadth or amount of variation in $p(\mathcal{M})$ itself. Intuitively, if the amount of diversity in the training set is small (i.e. low values of $n_{\text{train}}$) relative to the breadth of $p(\mathcal{M})$, then this poses a more difficult generalization challenge. As $n_{\text{train}} \to \infty$, we will eventually enter a regime where $p(\mathcal{M})$ is densely sampled, and generalization should become much easier.

Given the above definition of generalization, we distinguish between two qualitative types of distributions over MDPs. In *procedural* generalization, all MDPs with non-zero probability under $p(\mathcal{M})$ share the same underlying logic to their dynamics (e.g., that walls are impassable) and rewards (e.g., that coins are rewarding), but differ in how the environment is laid out (e.g., different mazes) and in how it is rendered (e.g., color or background). In *task* generalization, MDPs under $p(\mathcal{M})$ share the same dynamics and rendering, but differ in the reward function (e.g., picking an object up vs. pushing it), which may be parameterized (e.g. specifying a goal location).

---

[1]Technically, the different tasks contained in Meta-World (see Section 3) are not quite drawn IID as they were designed by hand. However, they qualitatively involve the same types of objects and level of complexity, so we feel this fits approximately into this regime.

**Procedural generalization**    Many recent works attempt to improve procedural generalization in different ways. For example, techniques that have been successful in supervised learning have also been shown to help procedural generalization in RL, including regularization [9, 33, 42] and auxiliary self-supervised objectives [43, 44]. Other approaches including better hyperparameter tuning [47], curriculum learning strategies [35], and stronger architectural inductive biases [5, 23, 75] may also be effective. However, these various ideas are often only explored in the model-free setting, and are not usually evaluated in combination, making it challenging to know which are the most beneficial for model-based RL. Here we focus explicitly on model-based agents, and evaluate three factors in combination: planning, self-supervision, and data diversity.

**Task generalization**    Generalizing to new tasks or reward functions has been a major focus of meta-RL [19, 57], in which an agent is trained on a dense distribution of tasks during training and a meta-objective encourages few-shot generalization to new tasks. However, meta-RL works have primarily focused on model-free algorithms [though see 48], and the distribution from which train/test MDPs are drawn is typically quite narrow and low dimensional. Another approach to task generalization has been to first train task-agnostic representations [72] or dynamics [64] in an exploratory pre-training phase, and then use them for transfer. Among these, Sekar et al. [64] focus on the model-based setting but require access to the reward function during evaluation. In Section 4.2, we take a similar approach of pre-training models and using them for task transfer, with the goal of evaluating whether planning and self-supervised learning might assist in this.

## 2.2    FACTORS OF GENERALIZATION

**Planning**    Model-based RL is an active area of research [28, 46] with the majority of work focusing on gains in data efficiency [26, 27, 34, 37, 61], though it is often motivated by a desire for better zero- or few-shot generalization as well [18, 29, 48, 64, 70]. In particular, model-based techniques may benefit data efficiency and generalization in three distinct ways. First, model learning can act as an auxiliary task and thus aid in learning representations that better capture the structure of the environment and enable faster learning [20]. Second, the learned model can be used to select actions on-the-fly via MPC [18], enabling faster adaptation in the face of novelty. Third, the model can also be used to train a policy or value function by simulating training data [66] or constructing more informative losses [21, 30], again enabling faster learning (possibly entirely in simulation). A recent state-of-the-art agent, MuZero [60], combines all of these techniques: model learning, MPC, simulated training data, and model-based losses.[2] We focus our analysis on MuZero for this reason, and compare it to baselines that do not incorporate model-based components.

**Self-supervision**    Model learning is itself a form of self-supervision, leveraging an assumption about the structure of MDPs in order to extract further learning signal from collected data than is possible via rewards alone. However, recent work has argued that this is unnecessary for model-based RL: all that should be required is for models to learn the task dynamics, not necessarily environment dynamics [22]. Yet even in the context of model-free RL, exploiting the structure of the dynamics has been shown to manifest in better learning efficiency, generalization, and representation learning [44, 63, 72]. Here, we aim to test the impact of self-supervision in model-based agents by focusing on three popular classes of self-supervised losses: reconstruction, contrastive, and self-predictive. Reconstruction losses involve directly predicting future observations [e.g. 18, 27, 37, 70]. Contrastive objectives set up a classification task to determine whether a future frame could result from the current observation [4, 38, 52]. Finally, self-predictive losses involve having agents predict their own future latent states [25, 63].

**Data diversity**    Several works have shown that exposing a deep neural network to a diverse training distribution can help its representations better generalize to unseen situations [16, 54]. A dense sampling over a diverse training distribution can also act as a way to sidestep the out-of-distribution generalization problem [54]. In robotics, collecting and training over a diverse range of data has been found to be critical particularly in the sim2real literature where the goal is to transfer a policy

---

[2]In this paper, we refer to *planning* as some combination of MPC, simulated training data, and model-based losses (excluding model learning iteself). We are agnostic to the question of how deep or far into the future the model must be used; indeed, it may be the case that in the environments we test, very shallow or even single-step planning may be sufficient, as in Hamrick et al. [29], Hessel et al. [31].

learned in simulation to the real world [3, 67]. Guez et al. [23] also showed that in the context of zero-shot generalization, the amount of data diversity can have interesting interactions with other architectural choices such as model size. Here, we take a cue from these findings and explore how important procedural data diversity is in the context of self-supervised, model-based agents.

## 2.3 MuZero

We evaluate generalization with respect to a state-of-the-art model-based agent, MuZero [60]. MuZero is an appealing candidate for investigating generalization because it already incorporates many ideas that are thought to be useful for generalization and transfer, including replay [61], planning [65], and model-based representation learning [22, 51]. However, while these components contribute to strong performance across a wide range of domains, other work has suggested that MuZero does not necessarily achieve perfect generalization on its own [29]. It therefore serves as a strong baseline but with clear room for improvement.

MuZero learns an implicit (or value-equivalent, see Grimm et al. [22]) world model by simply learning to predict future rewards, values and actions. It then plans with Monte-Carlo Tree Search (MCTS) [40, 11] over this learned model to select actions for the environment. Data collected from the environment, as well as the results of planning, are then further used to improve the learned reward function, value function, and policy. More specifically, MuZero optimizes the following loss at every time-step $t$, applied to a model that is unrolled $0 \ldots K$ steps into the future: $l_t(\theta) = \sum_{k=0}^{K}(l_\pi^k + l_v^k + l_r^k) = \sum_{k=0}^{K}(\text{CE}(\hat{\pi}^k, \pi^k) + \text{CE}(\hat{v}^k, v^k) + \text{CE}(\hat{r}^k, r^k))$, where $\hat{\pi}^k$, $\hat{v}^k$ and $\hat{r}^k$ are respectively the policy, value and reward prediction produced by the $k$-step unrolled model. The targets for these predictions are drawn from the corresponding time-step $t + k$ of the real trajectory: $\pi^k$ is the improved policy generated by the search tree, $v^k$ is an $n$-step return bootstrapped by a target network, and $r^k$ is the true reward. As MuZero uses a distributional approach for training the value and reward functions [13], their losses involve computing the cross-entropy (CE); this also typically results in better representation learning. For all our experiments, we specifically parameterize $v$ and $r$ as categorical distributions similar to the Atari experiments in Schrittwieser et al. [60].

To enable better sample reuse and improved data efficiency, we use the *Reanalyse* version of MuZero [61]. Reanalyse works by continuously re-running MCTS on existing data points, thus computing new improved training targets for the policy and value function. It does not change the loss function described in Section 2.3. In domains that use continuous actions (such as Meta-World), we use Sampled MuZero [32] that modifies MuZero to plan over sampled actions instead. See Section A.1 for more details on MuZero.

## 3 Experimental Design

We analyze three potential drivers of generalization (planning, self-supervision, and data diversity) across two different environments. For each algorithmic choice, we ask: to what extent does it improve procedural generalization, and to what extent does it improve task generalization?

### 3.1 Environments

**Procgen** Procgen [9] is a suite of 16 different Atari-like games with procedural environments (e.g. game maps, terrain, and backgrounds), and was explicitly designed as a setting in which to test procedural generalization. It has also been extensively benchmarked [9, 10, 35, 55]. For each game, Procgen allows choosing between two difficulty settings (easy or hard) as well as the number of levels seen during training. In our experiments, we use the "hard" difficulty setting and vary the numbers of training levels to test data diversity (see below). For each Procgen game, we train an agent for 30M environment frames. We use the same network architecture and hyper-parameters that Schrittwieser et al. [61] used for Atari and perform no environment specific tuning. We report mean normalized scores across all games as in Cobbe et al. [9]. Following the recommendation of Agarwal et al. [1], we report the min-max normalized scores across all games instead of PPO-normalized scores. Thus, the normalized score for each game is computed as $\frac{(score-min)}{(max-min)}$, where $min$ and $max$ are the minimum and maximum scores possible per game as reported in [9]. We then average the normalized scores across all games to report the mean normalized score.

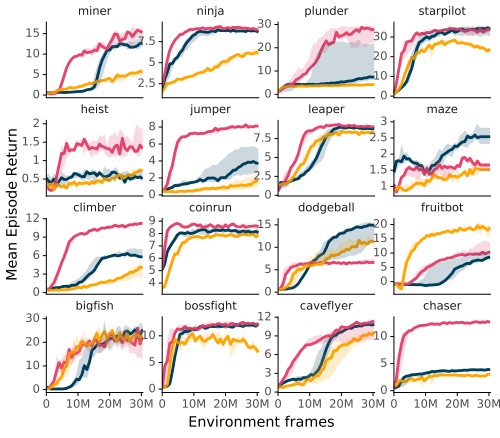 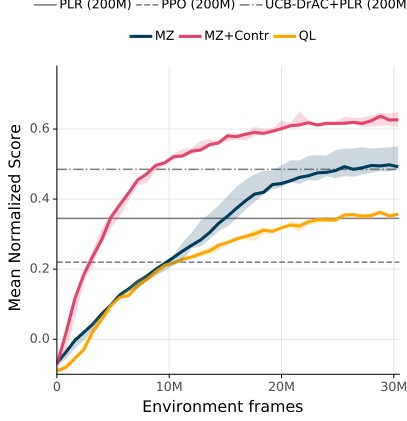

(a) Individual scores across all 16 Procgen games      (b) Mean normalized Score

Figure 2: The impact of planning and self-supervision on procedural generalization in Procgen (hard difficulty, 500 train levels). We plot the zero-shot evaluation performance on unseen levels for each agent throughout training. The Q-Learning agent (QL) is a replica of the MuZero (MZ) with its model-based components removed. MZ+Contr is a MuZero agent augmented with a temporal contrastive self-supervised loss that is action-conditioned (we study other losses in Figure 3). We observe that both planning and self-supervision improve procedural generalization on Procgen. Comparing with existing state-of-the-art methods which were trained for 200M frames on the right (PPO [62], PLR [35], and UCB-DrAC+PLR [56, 35], data from [35]), we note that MuZero itself exceeds state-of-the-art performance after being trained on only 30M frames. For all plots, dark lines indicate median performance across 3 seeds and the shaded regions denote the min and max performance across seeds. For training curves see Figure A.4, for additional metrics see Figure A.1.

**Meta-World**   Meta-World [74] is a suite of 50 different tasks on a robotic SAWYER arm, making it more suitable to test task generalization. Meta-World has three benchmarks focused on generalization: ML-1, ML-10, and ML-45. ML-1 consists of 3 goal-conditioned tasks where the objective is to generalize to unseen goals during test time. ML-10 and ML-45 require generalization to completely new tasks (reward functions) at test time after training on either 10 or 45 tasks, respectively. Meta-World exposes both state and pixel observations, as well as dense and sparse versions of the reward functions. In our experiments, we use the v2 version of Meta-World, dense rewards, and the `corner3` camera angle for pixel observations. For Meta-World, we trained Sampled MuZero [32] for 50M environment frames. We measure performance in terms of average episodic success rate across task(s). Note that this measure is different from task rewards, which are dense.

## 3.2   FACTORS OF GENERALIZATION

**Planning**   To evaluate the contribution of planning (see Section 2.2), we compared the performance of a vanilla MuZero Reanalyse agent with a Q-Learning agent. We designed the Q-Learning agent to be as similar to MuZero as possible: for example, it shares the same codebase, network architecture, and replay strategy. The primary difference is that the Q-Learning agent uses a Q-learning loss instead of the MuZero loss in Section 2.3 to train the agent, and uses $\epsilon$-greedy instead of MCTS to act in the environment. See Section A.2 for further details.

**Self-supervision**   We looked at three self-supervised methods as auxiliary losses on top of MuZero: image reconstruction, contrastive learning [24], and self-predictive representations [63]. We do not leverage any domain-specific data-augmentations for these self-supervised methods.

*Reconstruction.* Our approach for image reconstruction differs slightly from the typical use of mean reconstruction error over all pixels. In this typical setting, the use of averaging implies that the decoder can focus on the easy to model parts of the observation and still perform well. To encourage the decoder to model all including the hardest to reconstruct pixels, we add an additional loss term which corresponds to the max reconstruction error over all pixels. See Section A.3 for details.

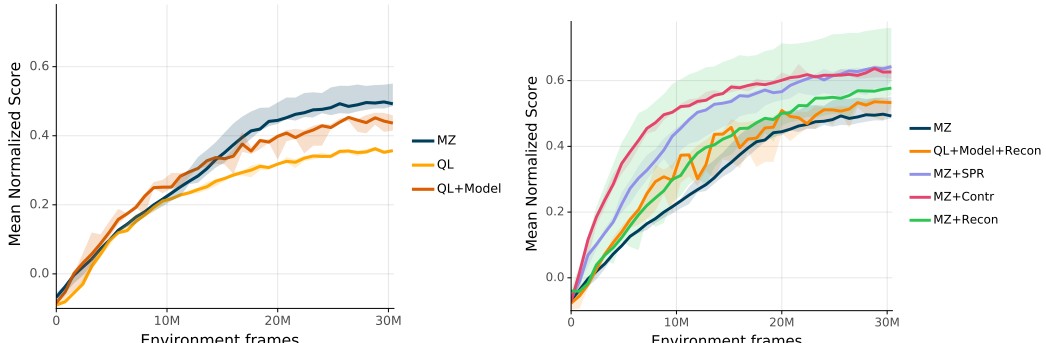

Figure 3: Evaluation performance on Procgen (hard, 500 train levels). On the left, we ablate the effectiveness of planning. The Q-Learning agent (QL) is a replica of MuZero (MZ) without model-based components. We then add a model to this agent (QL+Model) (see Section A.2) to disentangle the effects of the model-based representation learning from planning in the full MuZero model (MZ). On the right, we ablate the effect of self-supervision with three different losses: Contrastive (Contr), Self-Predictive (SPR), and Image Reconstruction (Recon). We also include a Q-Learning+Model agent with reconstruction (QL+Model+Recon) as a baseline. For all plots, dark lines indicate median performance across 3 seeds (5 seeds for MZ and MZ+Recon) and the shaded regions denote the min and max performance across seeds. For corresponding training curves see Figure A.5.

*Contrastive.* We also experiment with a temporal contrastive objective which treats pairs of observations close in time as positive examples and un-correlated timestamps as negative examples [2, 52, 23]. Our implementation of the contrastive objective is action-conditioned and uses the MuZero dynamics model to predict future embeddings. Similar to ReLIC [45], we also use a KL regularization term in addition to the contrastive loss. The contrastive loss operates entirely in the latent space and does not need decoding to the pixel space at each prediction step, unlike the reconstruction loss. See Section A.4 for details.

*Self-predictive.* Finally, we experiment with self-predictive representations (SPR) [63] which are trained by predicting future representations of the agent itself from a target network using the MuZero dynamics model. Similar to the contrastive loss, this objective operates in the latent space but does not need negative examples. See Section A.5 for details.

**Data diversity** To evaluate the contribution of data diversity (see Section 2.1), we ran experiments on Procgen in which we varied the number of levels seen during training (either 10, 100, 500, or $\infty$). In all cases, we evaluated on the infinite test split. Meta-World does not expose a way to modify the amount of data diversity, therefore we did not analyze this factor in our Meta-World experiments.

## 4 RESULTS

We ran all experiments according to the design described in Section 3. Unless otherwise specified, all reported results are on the test environments of Procgen and Meta-World (results on the training environments are also reported in the Appendix) and are computed as medians across seeds.

### 4.1 PROCEDURAL GENERALIZATION

Overall results on Procgen are shown in Figure 2 and Table A.2. MuZero achieves a mean-normalized test score of 0.50, which is slightly higher than earlier state-of-the-art methods like UCB-DrAC+PLR which was specifically designed for procedural generalization tasks [35], while also being much more data efficient (30M frames vs. 200M frames). MuZero also outperforms our Q-Learning baseline which gets a score of 0.36. Performance is further improved by the addition of self-supervision, indicating that both planning and self-supervised model learning are important.

**Effect of planning** While not reaching the same level of performance of MuZero, the Q-Learning baseline performs quite well and achieves a mean normalized score of 0.36, matching performance

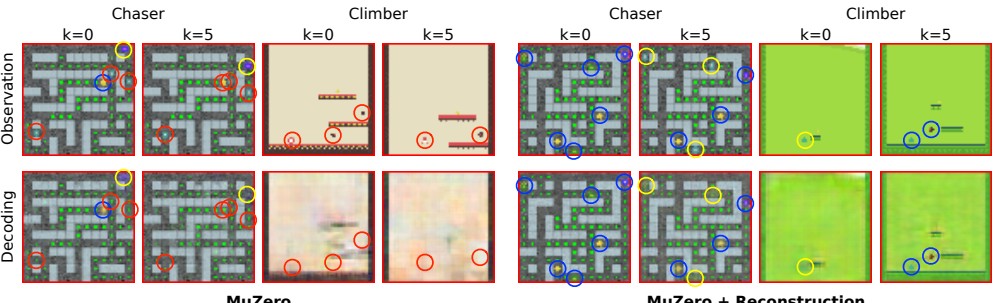

Figure 4: Qualitative comparison of the information encoded in the embeddings learned by MuZero with and without the auxiliary pixel reconstruction loss. For MuZero, embeddings are visualized by learning a standalone pixel decoder trained with MSE. Visualized are environment frames (top row) and decoded frames (bottom row) for two games (Chaser and Climber), for embeddings at the current time step ($k = 0$) and 5 steps into the future ($k = 5$). Colored circles highlight important entities that are or are not well captured (blue=captured, yellow=so-so, red=missing).

of other specialized model-free methods such as PLR [35]. By modifying the Q-Learning baseline to learn a 5-step value equivalent model (similar to MuZero), we find its performance further improves to 0.45, though does not quite catch up to MuZero itself (see Figure 3, left). This suggests that while simply learning a value-equivalent model can bring representational benefits, the best results come from also using this model for action selection and/or policy optimization.

We also tested the effect of planning in the Meta-World ML-1 benchmark from states. Table A.5 and Figure A.7 show the results. We find that both the Q-Learning and MuZero agents achieve perfect or near-perfect generalization performance on this task, although MuZero is somewhat more stable and data-efficient. The improvement of MuZero in stability or data efficiency again suggests that model-learning and planning can play a role in improving performance.

**Effect of self-supervision** Next, we looked at how well various self-supervised auxiliary losses improve over MuZero's value equivalent model on Procgen (Figure 3, right). We find that contrastive learning and self-predictive representations both substantially improve over MuZero's normalized score of 0.50 to 0.63 (contrastive) or 0.64 (SPR), which are new state-of-the-art scores on Procgen. The reconstruction loss also provides benefits but of a lesser magnitude, improving MuZero's performance from 0.50 to 0.57. All three self-supervised losses also improve the data efficiency of generalization. Of note is the fact that a MuZero agent with the contrastive loss can match the final performance of the baseline MuZero agent using only a third of the data (10M environment frames).

To further tease apart the difference between MuZero with and without self-supervision, we performed a qualitative comparison between reconstructed observations of MuZero with and without image reconstruction. The vanilla MuZero agent was modified to include image reconstruction, but with a stop gradient on the embedding so that the reconstructions could not influence the learned representations. We trained each agent on two games, Chaser and Climber. As can be seen in Figure 4, it appears that the primary benefit brought by self-supervision is in learning a more accurate world model and capturing more fine-grained details such as the position of the characters and enemies. In Chaser, for example, the model augmented with a reconstruction loss is able to to predict the position of the character across multiple time steps, while MuZero's embedding only retains precise information about the character at the current time step. This is somewhat surprising in light of the arguments made for value-equivalent models [22], where the idea is that world models do not need to capture full environment dynamics—only what is needed for the task. Our results suggest that in more complex, procedurally generated environments, it may be challenging to learn even the task dynamics from reward alone without leveraging environment dynamics, too. Further gains in model quality might be gained by also properly handling stochasticity [53] and causality [58].

**Effect of data diversity** Overall, we find that increased data diversity improves procedural generalization in Procgen, as shown in Figure 5. Specifically, at 10 levels, self-supervision barely improves the test performance on Procgen over MuZero. But as we increase the number of levels to

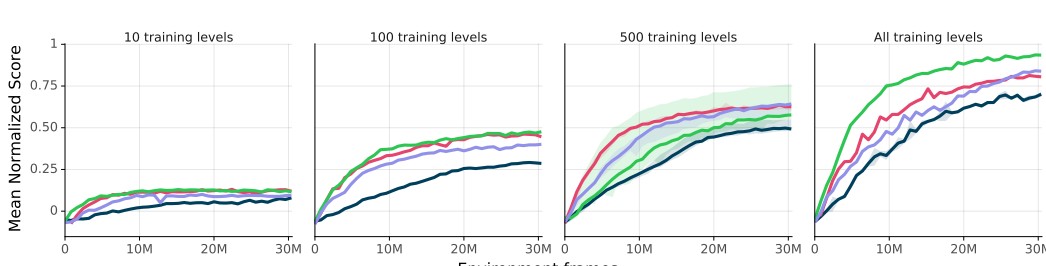

Figure 5: Interaction of self-supervision and data diversity on procedural generalization. Each plot shows generalization performance as a function of environment frames for different numbers of training levels. With only 10 levels, self-supervision does not bring much benefit over vanilla MuZero. Once the training set includes at least 100 levels, there is large improvement with self-supervised learning both in terms of data efficiency and final performance. For all plots, dark lines indicate median performance across seeds and shading indicates min/max seeds. See also Figure A.6 for corresponding training curves.

100, 500, or $\infty$, we observe a substantial improvement when using self-supervision. This is an interesting finding which suggests that methods that improve generalization might show promise only when we evaluate them on environments with a lot of inherent diversity, such as procedurally generated environments. Consequently, evaluating methods solely on single task settings (as is common in RL) might lead us to overlook innovations which might have a big impact on generalization.

Not only do we find that MuZero with self-supervision achieves better final generalization performance than the baselines, we also observe that self-supervision improves generalization performance *even when controlling for training performance*. To see this, we visualized generalization performance as a function of training performance (see Figure A.3). The figure shows that even when two agents are equally strong (according to the rewards achieved during training), they differ at test time, with those trained with self-supervision generally achieving stronger generalization performance. This suggests that in addition to improving data efficiency, self-supervision leads to more robust representations—a feature that again might be overlooked if not measuring generalization.

## 4.2 TASK GENERALIZATION

As we have shown, planning, self-supervision, and data diversity all play an important role in procedural generalization. We next investigated whether this trend holds for task generalization, too. To test this, we pre-trained the Q-Learning and MuZero agents with and without self-supervision on the ML-10 and ML-45 training sets of Meta-World and then evaluated zero-shot test performance. In all experiments, we trained agents from pixels using the dense setting of the reward functions. We report the agent's success rate computed the same way as in Yu et al. [74].

|  |  | MAML | RL$^2$ | QL | MZ | MZ+Recon |
|---|---|---|---|---|---|---|
| ML-10 | train | 44.4% | 86.9% | 85.2% | 97.6% | **97.8%** |
|  | zero-shot | - | - | 6.8% | **26.5%** | 25.0% |
|  | few-shot | 31.6% | **35.8%** | - | - | - |
|  | finetune | - | - | 80.1% | 94.1% | **97.5%** |
| ML-45 | train | 40.7% | 70% | 55.9% | **77.2%** | 74.9% |
|  | zero-shot | - | - | 10.8% | 17.7% | **18.5%** |
|  | few-shot | **39.9%** | 33.3% | - | - | - |
|  | finetune | - | - | 78.1% | 76.7% | **81.7%** |

Table 1: Train and various test success rates (zero-shot, few-shot, finetuning) on the ML-10 and ML-45 task generalization benchmarks of Meta-World. Shown are baseline results on MAML [19] and RL$^2$ [17] from the Meta-World paper [73] as well as our results with Q-Learning and MuZero. Note that MAML and RL$^2$ were trained for around 400M environment steps from states, whereas MuZero was trained from pixels for 50M steps on train tasks and 10M steps on test tasks for fine-tuning.

As shown in Table 1, we can observe that MuZero reaches better training performance on ML-10 (97.6%) and ML-45 (77.2%) compared to existing state-of-the-art agents [74]. These existing approaches are meta-learning methods, and are therefore not directly comparable in terms of test performance due to different data budgets allowed at test time; although MuZero does not reach the same level of performance as these agents at test, we find it compelling that it succeeds as often as it does, especially after being trained for only 50M environment steps (compared to 400M for the

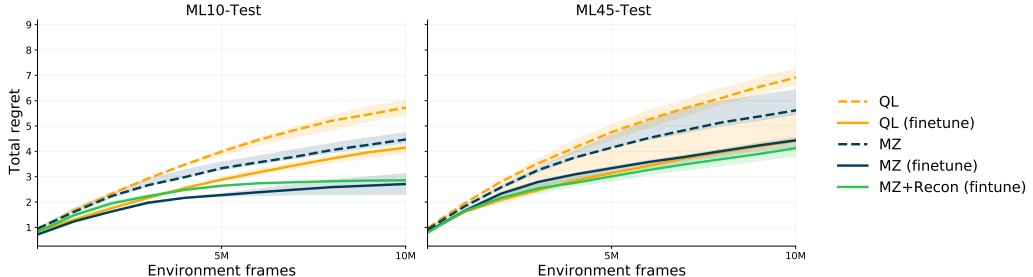

Figure 6: Finetuning performance on ML-10 and ML-45, shown as cumulative regret over the success rate (**lower is better**). Both the pre-trained Q-Learning and MuZero agents have lower regret than corresponding agents trained from scratch. MuZero also achieves lower regret than Q-Learning, indicating a positive benefit of planning (though the difference is small on ML-45). Self-supervision (pixel reconstruction) does not provide any additional benefits. Solid lines indicate median performance across seeds, and shading indicates min/max seeds. See also Figure A.8.

baselines). MuZero also outperforms Q-Learning in terms of zero-shot test performance, indicating a positive benefit of planning for task generalization. However, reconstruction and other forms of self-supervision do not improve performance and may even decrease it (see also Figure A.8).

We also looked at whether the pre-trained representations or dynamics would assist in more data-efficient task transfer by fine-tuning the agents on the test tasks for 10M environment steps. Figure 6 shows the results, measured as cumulative regret (see also Figure A.8 for learning curves). Using pre-trained representations or dynamics does enable both Q-Learning and MuZero to outperform corresponding agents trained from scratch—evidence for weak positive transfer to unseen tasks for these agents. Additionally, MuZero exhibits better data efficiency than Q-Learning, again showing a benefit for planning. However, self-supervision again does not yield improvements in fine-tuning, and as before may hurt in some cases (Figure A.8). This indicates that while MuZero (with or without self-supervision) excels at representing variation across tasks seen during training, there is room for improvement to better transfer this knowledge to unseen reward functions.

We hypothesize that MuZero only exhibits weak positive transfer to unseen tasks due to a combination of factors. First, the data that its model is trained on is biased towards the training tasks, and thus may not be sufficient for learning a globally-accurate world model that is suitable for planning in different tasks. Incorporating a more exploratory pre-training phase [e.g. 64] might help to alleviate this problem. Second, because the model relies on task-specific gradients, the model may over-represent features that are important to the training tasks (perhaps suggesting that value-equivalent models [22] may be poorly suited to task generalization). Third, during finetuning, the agent must still discover what the reward function is, even if it already knows the dynamics. It is possible that the exploration required to do this is the primary bottleneck for task transfer, rather than the model representation itself.

## 5    CONCLUSION

In this paper, we systematically investigate how well modern model-based methods perform at hard generalization problems, and whether self-supervision can improve the generalization performance of such methods. We find that in the case of procedural generalization in Procgen, both model-based learning and self-supervision have additive benefits and result in state-of-the-art performance on test levels with remarkable data efficiency. In the case of task generalization in Meta-World, we find that while a model-based agent does exhibit weak positive transfer to unseen tasks, auxiliary self-supervision does not provide any additional benefit, suggesting that having access to a good world model is not always sufficient for good generalization [29]. Indeed, we suspect that to succeed at task generalization, model-based methods must be supplemented with more sophisticated online exploration strategies, such as those learned via meta-learning [17, 39, 68] or by leveraging world models in other ways [64]. Overall, we conclude that self-supervised model-based methods are a promising starting point for developing agents that generalize better, particularly when trained in rich, procedural, and multi-task environments.

## REPRODUCIBILITY STATEMENT

In this paper, we have taken multiple steps to ensure that the algorithms, experiments, and results are as reproducible as possible. Both environments used in our experiments, Procgen and Meta-World, are publicly available. In order to produce reliable results, we ran multiple seeds in our key experiments. Our choice of agent is based on MuZero, which is published work [32, 60, 61]. In addition, we have included an extensive appendix (Appendix A) which describes the architectural details of MuZero, the exact hyperparameters used for all experiments, and a detailed description of the Q-Learning agents and the self-supervised losses.

## ACKNOWLEDGEMENTS

We would like to thank Jovana Mitrović, Ivo Danihelka, Feryal Behbahani, Abe Friesen, Peter Battaglia and many others for helpful suggestions and feedback on this work.

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

# A    Agent Details

## A.1    MuZero

**Network Architecture:**    We follow the same network architectures follow the ones used in MuZero ReAnalyse [61]. For pixel based inputs, the images are first sent through a convolutional stack that downsamples the $96 \times 96$ image down to an $8 \times 8$ tensor (for Procgen) or a $16 \times 16$ tensor (for Meta-World). This tensor then serves as input to the encoder. Both the encoder and the dynamics model were implemented by a ResNet with 10 blocks, each block containing 2 layers. For pixel-based inputs (Procgen and ML-45) each layer of the residual stack was convolutional with a kernel size of 3x3 and 256 planes. For state based inputs (ML-1) each layer was fully-connected with a hidden size of 512.

**Hyperparameters**    We list major hyper-parameters used in this work in Table A.1.

Table A.1: Hyper-parameters

| Hyper-parameter | Value (Procgen) | Value (Meta-World) |
|---|---|---|
| **Training** | | |
| Model Unroll Length | 5 | 5 |
| TD-Steps | 5 | 5 |
| ReAnalyse Fraction | 0.945 | 0.95 |
| Replay Size (in sequences) | 50000 | 2000 |
| **MCTS** | | |
| Number of Simulations | 50 | 50 |
| UCB-constant | 1.25 | 1.25 |
| Number of Samples | n/a | 20 |
| **Self-Supervision** | | |
| Reconstruction Loss Weight | 1.0 | 1.0 |
| Contrastive Loss Weight | 1.0 | 0.1 |
| SPR Loss Weight | 10.0 | 1.0 |
| **Optimization** | | |
| Optimizer | Adam | Adam |
| Initial Learning Rate | $10^{-4}$ | $10^{-4}$ |
| Batch Size | 1024 | 1024 |

**Additional Implementation Details:**    For Meta-World experiments, we provide the agent with past reward history as well. We found this to be particularly helpful when training on Meta-World since implicit task inference becomes easier. In both Procgen and Meta-World, the agent is given a history of the 15 last observations. The images are concatenated by channel and then input as one tensor to the encoder. For each game, we trained our model using 2 TPUv3-8 machines. A separate actor gathered environment trajectories with 1 TPUv3-8 machine.

## A.2    Controlled Model-free Baseline

Our controlled Q-Learning baseline begins with the same setup as the MuZero agent (Section A.1) and modifies it in a few key ways to make it model-free rather than model-based.

The Q-Learning baseline uses $n$-step targets for action value function. Given a trajectory $\{s_t, a_t, r_t\}_{t=0}^{T}$, the target action value is computed as follow

$$Q_{target}(s_t, a_t) = \sum_{i=0}^{n-1} \gamma^i r_{t+i} + \gamma^n \max_{a \in \mathcal{A}} Q_\xi(s_{t+n}, a) \qquad (1)$$

Where $Q_\xi$ is the target network whose parameter $\xi$ is updated every 100 training steps.

In order to make the model architecture most similar to what is used in the MuZero agent, we decompose the action value function into two parts: a reward prediction $\hat{r}$ and a value prediction $V$, and model these two parts separately. The total loss function is, therefore, $\mathcal{L}_{total} = \mathcal{L}_{reward} + \mathcal{L}_{value}$. The reward loss is exactly the same as that of MuZero. For the value loss, we can decompose Equation 1 in the same way:

$$
\begin{aligned}
Q_{target}(s_t, a_t) &= \hat{r}_t + \gamma V_{target}(s) \\
&= \sum_{i=0}^{n-1} \gamma^i r_{t+i} + \gamma^n \max_{a \in \mathcal{A}} \left( \hat{r}_{t+n} + \gamma V_\xi(s') \right) \\
\implies V_{target}(s) &= \sum_{i=1}^{n-1} \gamma^{i-1} r_{t+i} + \gamma^{n-1} \max_{a \in \mathcal{A}} \left( \hat{r}_{t+n} + \gamma V_\xi(s') \right)
\end{aligned}
\tag{2}
$$

Since the reward prediction should be taken care of by $\mathcal{L}_{reward}$ and it usually converges fast, we assume $\hat{r}_t = r_t$ and the target is simplified to Equation 2. We can then use this value target to compute the value loss $\mathcal{L}_{value} = \text{CE}(V_{target}(s), V(s))$.

In Meta-World, since it has a continuous action space, maximizing over the entire action space is infeasible. We follow the Sampled Muzero approach [32] and maximize only over the sampled actions.

The Q-Learning baseline uses 5-step targets for computing action values. This is the same as in MuZero, but unlike MuZero, the Q-Learning baseline only trains the dynamics for a single step. However, we also provide results for a 5-step dynamics function which we call QL+Model and QL+Model+Recon which further adds an auxiliary for reconstruction.

### A.3   MUZERO + RECONSTRUCTION

The decoder architecture mirrors the ResNet encoder with 10 blocks, but with each convolutional layer replaced by a de-convolutional layer [50]. In addition to the regular mean reconstruction loss, we add an additional max-loss term which computes the maximum reconstruction loss across all pixels. This incentives the reconstruction objective to not neglect the hardest to reconstruct pixels, and in turn makes the reconstructions sharper. The reconstruction loss can be used as follows:

$$
\mathcal{L}(X, Y) = \left( \frac{1}{|X|} \sum_{i,j,f} \ell(X_{ijf}, Y_{ijf}) \right) + \max_{i,j,f} \ell(X_{ijf}, Y_{ijf}),
$$

where $\ell(x, y) = (x - y)^2$ is the element-wise squared distance between each feature (i.e. RGB) value in the image array, where $i$ and $j$ are the pixel indices, where $f$ is the feature index, and where $|X|$ is the size of the array.

### A.4   MUZERO + CONTRASTIVE

Our implementation of the contrastive loss does not rely on any image augmentations but focuses on the task of predicting future embeddings of the image encoder [4, 52]. Howerver, in this case, the dynamics function produces a predicted vector $x_n$ at each step $n$ in the unroll, making the contrastive task action-conditional similar to CPC|Action [24]. We utilize the same target network used in train temporal difference in MuZero to compute our target vectors. We then attempt to match $x_n$ with the corresponding target vector $y_n$ directly computed from the observation at that timestep. We compute the target vector $y_n$ with the same target network weights used to compute the temporal difference loss in MuZero. Since the contrastive loss is ultimately a classification problem, we also need negative examples. For each pair $x_n$ and $y_n$, the sets $\{x_i\}_{i=0, i \neq n}^N$ and $\{y_i\}_{i=0, i \neq n}^N$ serve as the negative examples with $N$ the set of randomly sampled examples in the minibatch. We randomly sample 10% of the examples to constitute $N$. As in previous work [4, 6, 45] we use a cosine distance between $x_i$ and $y_i$ and feed this distance metric through a softmax function. We use the loss function in ReLIC [45] and CoBERL [4] and use a temperature of 0.1.

The 256-dimensional vector $x_n$ is derived from a two layer critic function which is first a convolutional layer with stride 1 and kernel size 3 on top of the state of the dynamics function. This is then fed into a fully connected layer to produce $x_n$. For each step $n$, the target vector $y_n$, which serves as a positive example, is derived from the final convolutional layer (8) that is fed into the residual stack of the encoder. The vector is computed by inputting the encoder, using the target network weights, with the corresponding image at that future step $n$. The encoding is then fed into the critic network to compute $y_n$.

The encoder of MuZero uses 15 past images as history. However, when we compute the target vectors, our treatment of the encoding history is different from that of the agent; instead of stacking all of the historical images ($n - 15...n - 1$) up to the corresponding step $n$, we simply replace the history stack with 15 copies of the image at the current step $n$. We found that this technique enhanced the performance of contrastive learning.

## A.5 MuZero + SPR

The implementation of SPR is similar to the contrastive objective except that it does not use negative examples. Thus it only uses a mean squared error instead of InfoNCE. The architecture for SPR is nearly identical to what is described in Section A.4. The same layers mentioned compute predicted and target vectors $x_n$ and $y_n$ respectively. However, SPR relies on an additional projection layer for learning. We thus add a projection layer $p_n$ of 256 dimensions on top of the critic function mentioned in Section A.4. Instead of computing a loss between $x_n$ and $y_n$, we follow the original formulation of SPR and compute mean squared error between L2-normalized $p_n$ and L2-normalized $y_n$.

# B  ADDITIONAL RESULTS

## B.1  COMPARISON OF Q-LEARNING AND MUZERO ON PROCGEN

As discussed in the main text, we trained a 5-step value-equivalent model on top of the Q-Learning agent. The performance of this agent with and without supervision, as well as the baseline Q-Learning agent, is reported in Table A.2 and visualized in Figure A.2. Both incorporating the 5-step model and adding reconstruction loss helps the Q-Learning agent's performance.

When using model-based planning, it's important to ensure that the model is "correct". Otherwise, planning on an "incorrect" model could hurts the agent performance, as we can see for the last few games in Figure A.2. Specifically, we can see that on these games, MuZero performs far worse than Q-Learning; yet, its performance is dramatically improved by the addition of the reconstruction loss. This suggests that without self-supervision, MuZero's model is poor, resulting in compounding model errors during planning. Figure 4 provides a qualitative analysis of MuZero's model in two of these games, confirming this hypothesis.

| Game | Mode | MuZero | | | | Q-Learning | | |
|------|------|--------|--------|--------|--------|------|--------|--------|
| | | MZ | MZ+Contr | MZ+Recon | MZ+SPR | QL | QL+Model | QL+Model+Recon |
| bigfish | train | 0.77 | 0.73 | 0.80 | 0.79 | 0.60 | 0.53 | 0.57 |
| | test | 0.60 | 0.55 | 0.61 | 0.59 | 0.58 | 0.59 | 0.51 |
| bossfight | train | 0.91 | 0.93 | 0.94 | 0.94 | 0.51 | 0.64 | 0.71 |
| | test | 0.92 | 0.94 | 0.92 | 0.95 | 0.57 | 0.65 | 0.79 |
| caveflyer | train | 0.91 | 0.90 | 0.87 | 0.67 | 0.82 | 0.83 | 0.88 |
| | test | 0.77 | 0.81 | 0.77 | 0.58 | 0.65 | 0.64 | 0.73 |
| chaser | train | 0.28 | 0.89 | 0.26 | 0.89 | 0.16 | 0.53 | 0.54 |
| | test | 0.24 | 0.89 | 0.24 | 0.89 | 0.17 | 0.68 | 0.65 |
| climber | train | 0.72 | 0.96 | 0.92 | 0.94 | 0.42 | 0.58 | 0.81 |
| | test | 0.42 | 0.87 | 0.76 | 0.83 | 0.27 | 0.40 | 0.68 |
| coinrun | train | 0.98 | 0.98 | 0.98 | 0.97 | 0.85 | 0.87 | 0.87 |
| | test | 0.64 | 0.71 | 0.68 | 0.73 | 0.58 | 0.60 | 0.68 |
| dodgeball | train | 0.78 | 0.30 | 0.74 | 0.42 | 0.58 | 0.60 | 0.62 |
| | test | 0.77 | 0.29 | 0.73 | 0.41 | 0.54 | 0.64 | 0.59 |
| fruitbot | train | 0.33 | 0.44 | 0.55 | 0.52 | 0.56 | 0.61 | 0.65 |
| | test | 0.31 | 0.38 | 0.57 | 0.48 | 0.69 | 0.72 | 0.75 |
| heist | train | 0.13 | 0.17 | 0.18 | 0.13 | 0.14 | 0.14 | 0.25 |
| | test | -0.17 | -0.09 | -0.18 | -0.16 | -0.17 | -0.17 | 0.05 |
| jumper | train | 0.75 | 0.89 | 0.87 | 0.90 | 0.54 | 0.61 | 0.82 |
| | test | 0.32 | 0.78 | 0.77 | 0.73 | 0.06 | 0.09 | 0.66 |
| leaper | train | 0.98 | 0.98 | 0.97 | 0.97 | 0.70 | 0.75 | 0.74 |
| | test | 0.86 | 0.90 | 0.88 | 0.90 | 0.80 | 0.85 | 0.85 |
| maze | train | 0.58 | 0.38 | 0.54 | 0.44 | 0.26 | 0.30 | 0.39 |
| | test | -0.25 | -0.39 | -0.19 | -0.31 | -0.41 | -0.39 | -0.39 |
| miner | train | 0.82 | 0.96 | 1.06 | 1.06 | 0.27 | 0.46 | 0.56 |
| | test | 0.59 | 0.78 | 0.85 | 0.94 | 0.23 | 0.15 | 0.11 |
| ninja | train | 0.98 | 0.98 | 0.98 | 0.98 | 0.70 | 0.77 | 0.83 |
| | test | 0.84 | 0.87 | 0.83 | 0.85 | 0.52 | 0.61 | 0.78 |
| plunder | train | 0.38 | 0.97 | 0.31 | 0.97 | 0.32 | 0.41 | 0.50 |
| | test | 0.17 | 0.94 | 0.13 | 0.91 | 0.04 | 0.05 | 0.40 |
| starpilot | train | 1.19 | 1.16 | 1.23 | 0.93 | 0.65 | 0.76 | 0.77 |
| | test | 0.98 | 0.95 | 0.91 | 0.84 | 0.66 | 0.77 | 0.80 |
| Average | train | 0.72 (0.70/0.75) | **0.78** (0.78/0.79) | 0.75 (0.74/0.88) | **0.79** (0.71/0.79) | 0.51 (0.49/0.51) | 0.60 (0.59/0.60) | 0.66 (0.58/0.66) |
| | test | 0.50 (0.48/0.55) | **0.63** (0.61/0.65) | 0.57 (0.54/0.75) | **0.64** (0.57/0.64) | 0.36 (0.35/0.37) | 0.45 (0.44/0.47) | 0.54 (0.45/0.54) |

Table A.2: Final normalised training and test scores on Procgen when training on 500 levels for 30M environment frames, reporting the median across 3 seeds (for MZ, MZ+Recon across 5 seeds). Min/max over seeds are shown in parentheses. Final scores for each seed were computed as means over data points from the last 1M frames. A subset of the experiments on Maze have terminated before 30M frames therefore we computed final scores at 27M for all agents on this game.

| Training levels | Mode | MZ | MZ+Contr | MZ+Recon | MZ+SPR |
|---|---|---|---|---|---|
| 10 | train | 0.73 | 0.76 | 0.76 | 0.68 |
| | test | 0.07 | 0.13 | 0.12 | 0.09 |
| 100 | train | 0.72 | 0.76 | 0.82 | 0.79 |
| | test | 0.29 | 0.46 | 0.47 | 0.40 |
| 500 | train | 0.72 | 0.78 | 0.75 | 0.79 |
| | test | 0.50 | 0.63 | 0.57 | 0.64 |
| All | train | 0.71 | 0.82 | 0.93 | 0.85 |
| | test | 0.68 | 0.80 | 0.93 | 0.83 |

Table A.3: MuZero with self-supervised losses on Procgen. Final mean normalised training and test scores for different number of training levels. Reporting median values over 1 seed on 10, 100 levels; on 500 levels 3 seeds for MZ+Contr, MZ+SPR and 5 seeds for MZ, MZ+Recon; on infinite levels 1 seed for MZ+Contr, MZ+SPR and 2 seeds for MZ, MZ+Recon. Final scores for each seed were computed as means over data points from the last 1M frames.

| Mode | PPO (200M) | PLR (200M) | UCB-DrAC+PLR (200M) |
|---|---|---|---|
| train | 0.41 | 0.53 | 0.61 |
| test | 0.220 | 0.345 | 0.485 |

Table A.4: Mean-normalized scores on ProcGen for PPO [62], PLR [35] and UCB-DraC+PLR ([56, 35]) used in Figure 2 and Figure A.4. For all scores in this table, we use the experimental data from Jiang et al. [35].

| | reach | push | pick-place |
|---|---|---|---|
| Agent | Test | Test | Test |
| RL$^2$ | 100% | 96.5% | 98.5% |
| MuZero | 100% | 100% | 100% |
| Q-Learning | 97.5% | 99.8% | 99.9% |

Table A.5: Zero-shot test performance on the ML-1 procedural generalization benchmark of Meta-World. Shown are baseline results on RL$^2$ [17] from the Meta-World paper [73] as well as our results on MuZero. Note that RL$^2$ is a meta-learning method and was trained for 300M environment steps, while MuZero and Q-Learning do not require meta-training were only trained for 50M steps and fine-tune for 10M. We average the success rate of the last 1M environment steps for each seed and report medians across three seeds for MuZero and Q-Learning (RL$^2$ results from [17]).

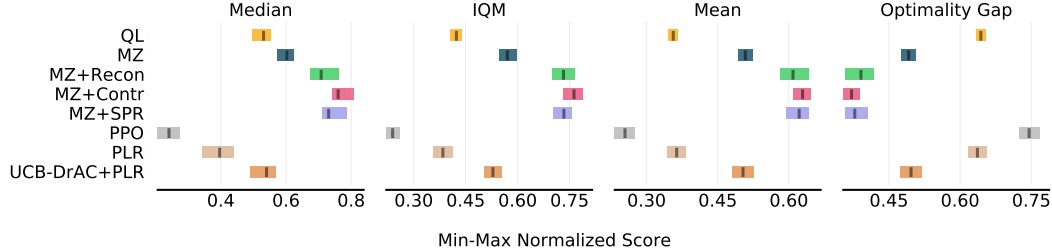

Figure A.1: Additional metrics (proposed in Agarwal et al. [1]) indicating zero-shot test performance of different methods on ProcGen. IQM corresponds to the Inter-Quratile Mean among all runs, and Optimality Gap refers to the amount by which the algorithm fails to meet a minimum score of 1.0.

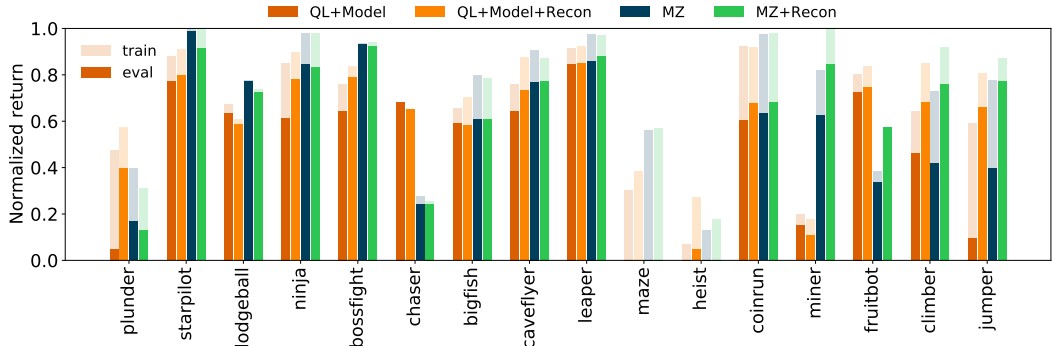

Figure A.2: Per-game breakdown of scores for MuZero and Q-Learning agents along with their reconstruction variants. Shaded bars indicate training performance while solid bars indicate zero-shot test performance. We average the normalized return of the last 1M environment steps for each seed and report medians across 5 seeds for MZ and MZ+Recon and 3 seeds for all other agents.

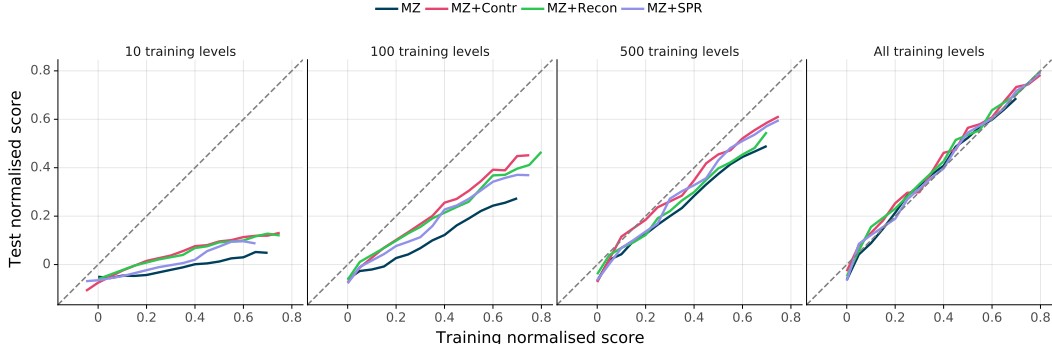

Figure A.3: Isolating the interaction between self-supervision and data diversity. Each plot shows the generalization performance as a function of training performance for different numbers of training levels. Also shown is the unity line, where training and testing performance are equivalent (note that with infinite training levels, all lines collapse to the unity line, as in this case there is no difference between train and test). Generally speaking, self-supervision results in stronger generalization than MuZero. Reporting median values over 1 seed on 10, 100 levels; on 500 levels 3 seeds for MZ+Contr, MZ+SPR and 5 seeds for MZ, MZ+Recon; on infinite levels 1 seed for MZ+Contr, MZ+SPR and 2 seeds for MZ, MZ+Recon. For visibility, min/max seeds are not shown.

| Game | Mode | MuZero | | | | Q-Learning |
| | | MZ | MZ+Contr | MZ+Recon | MZ+SPR | QL |
|---|---|---|---|---|---|---|
| ML10 | train | 97.6% | 78.1% | 97.8% | 82.5% | 85.2% |
| | zero-shot test | 26.5% | 17.0% | 25.0% | 17.8% | 6.8% |
| | fine-tune test | 94.1% | 60.0% | 97.5% | 51.3% | 80.1% |
| ML45 | train | 77.2% | 57.5% | 74.9% | 54.4% | 55.9% |
| | zero-shot test | 17.7% | 12.5% | 18.5% | 16.0% | 10.8% |
| | fine-tune test | 76.7% | 77.3% | 81.7% | 77.2% | 78.1% |

Table A.6: Success rates for MuZero (with and without self-supervision) and Q-Learning on ML-10 and ML-45. Training results are shown at 50M frames and fine-tuning results are shown at 10M frames. Reporting average of the last 1M frames and then medians across 3 seeds (2 for MZ and MZ+Recon on ML10).

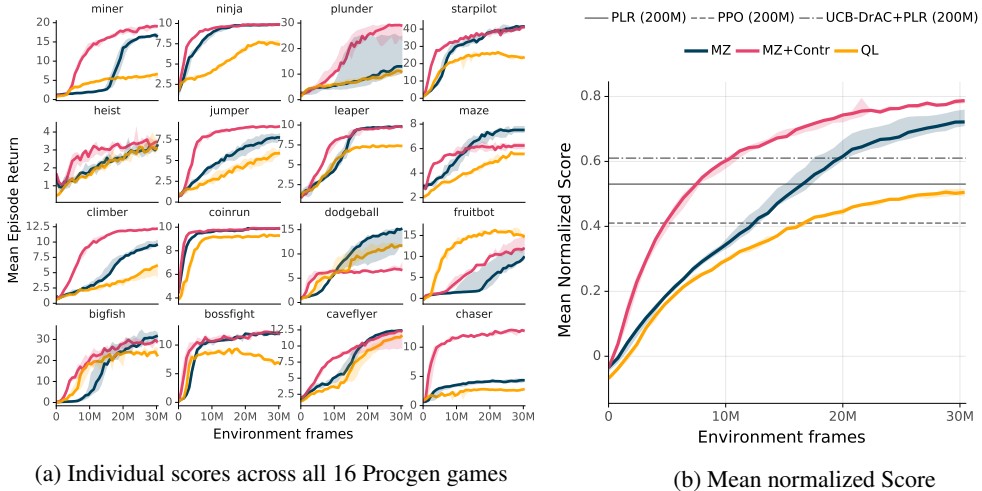

(a) Individual scores across all 16 Procgen games

(b) Mean normalized Score

Figure A.4: Training performance on Procgen. See Figure 2 in the main text for results on the test set. Reporting median values over 3 seeds (5 seeds for MZ), with shaded regions indicating min/max seeds.

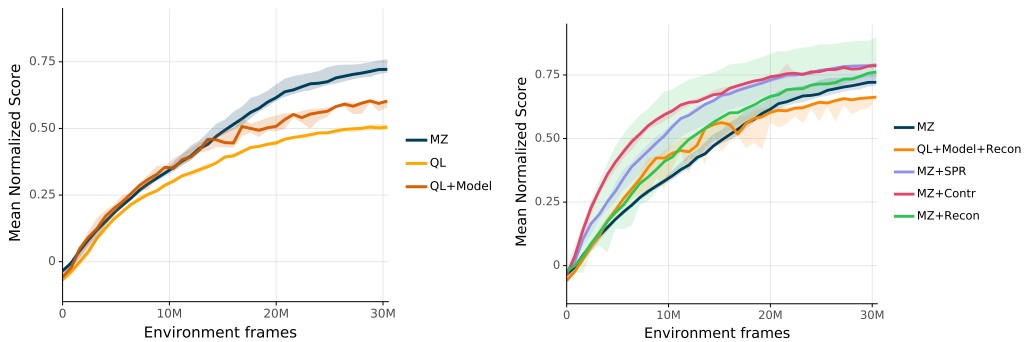

Figure A.5: Training performance on Procgen. See Figure 3 in the main text for corresponding results on the test set. Left: the effect of planning. Right: the effect of self-supervision. Reporting median performance over 3 seeds (5 seeds for MZ and MZ+Recon), with shaded regions indicating min/max seeds.

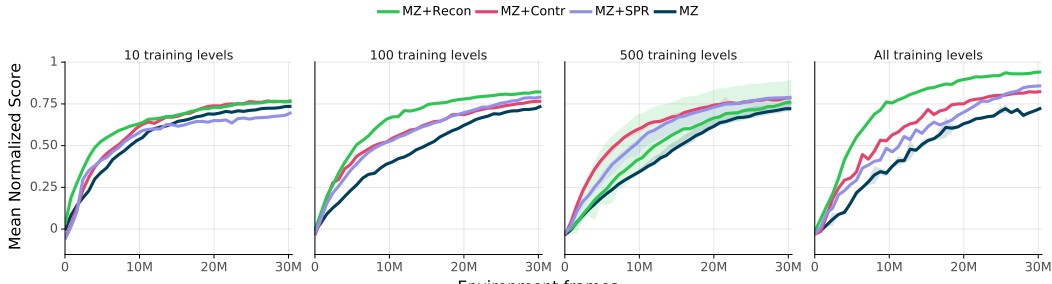

Figure A.6: Interaction of self-supervision and data diversity on training performance. See Figure 5 in the main text for corresponding results on the test set. Reporting median values over 1 seed on 10, 100 levels; on 500 levels 3 seeds for MZ+Contr, MZ+SPR and 5 seeds for MZ, MZ+Recon; on infinite levels 1 seed for MZ+Contr, MZ+SPR and 2 seeds for MZ, MZ+Recon. Shaded regions indicate min/max seeds.

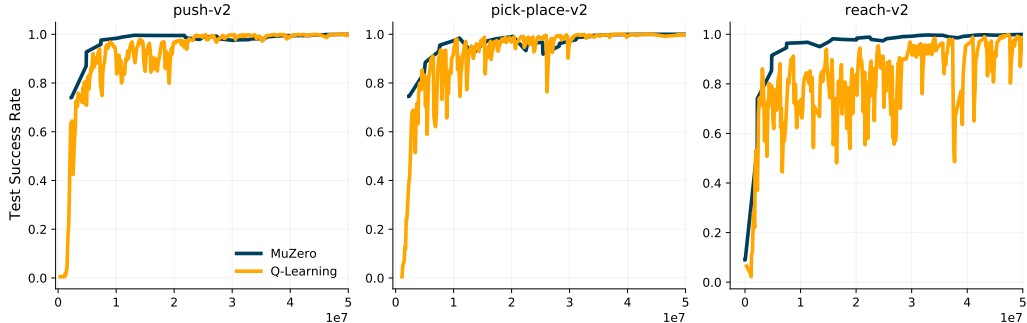

Figure A.7: Zero-shot test success rates MuZero and Q-Learning on unseen goals on the ML-1 procedural generalization benchmark of Meta-World. While both MuZero and Q-Learning achieve near optimal performance, MuZero is more stable and a bit more data-efficient than Q-Learning. Shown are medians across three seeds (for visibility, min/max seeds are not shown).

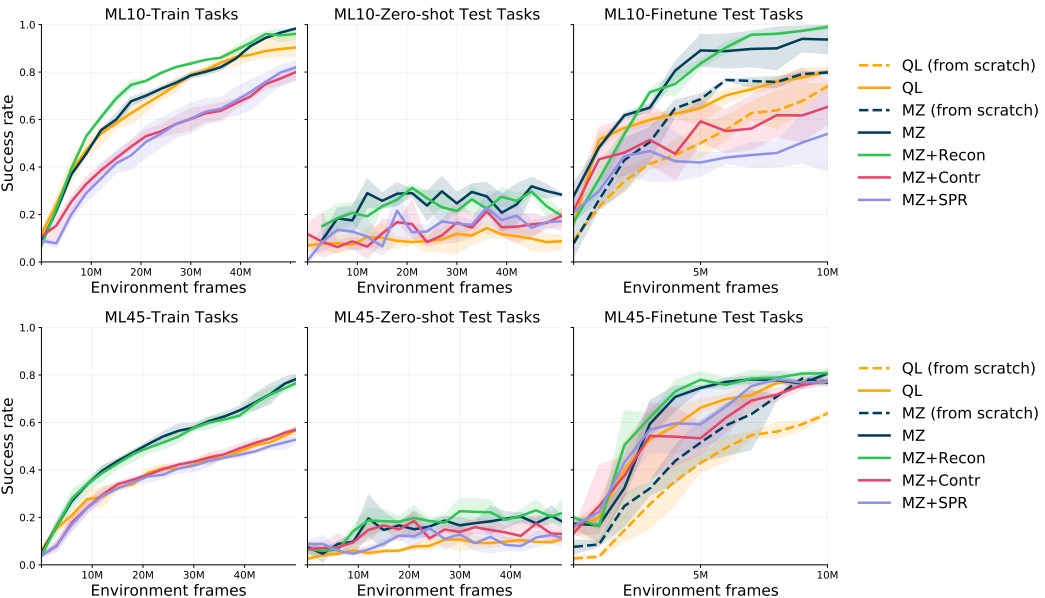

Figure A.8: Train, zero-shot test, and fine-tuning performance on ML-10 and ML-45. The top row shows ML-10, while the bottom row shows ML-45. The left column shows training performance, the middle column shows zero-shot test performance, and the right column shows fine-tuning performance. Reporting medians across 3 seeds (2 seeds for MZ and MZ+Recon on ML10), with shaded regions indicating min/max seeds. See also Figure 6 for plots showing cumulative regret during fine-tuning.

