# OpenReview forum: "Procedural generalization by planning with self-supervised world models"
_ICLR.cc/2022/Conference — ICLR 2022 Poster_

### Official Review · Reviewer_hXT2 · 2021-10-28

**Correctness:** 4
**Technical Novelty And Significance:** 1
**Empirical Novelty And Significance:** 3
**Recommendation:** 6
**Confidence:** 3

**Main Review:**

Strengths:

- The paper is very well written and the motivation and findings are clearly presented and easy to follow.
- They provide a good ablation study of the different losses and explored a few interesting cases.

Weaknesses:

- The main weakness of the paper lies in that the scope is somewhat limited to an empirical study with only a small number of take-away learnings for the community, which are not necessarily very surprising.
- Minor: “Although significant strides have been made in model-based systems in recent years [27], the most popular model-based benchmarks consist of identical training and testing environments [e.g. 26, 68] and do not measure or optimize for for generalization at all.” remove double “for”.


**Summary Of The Paper:**

The authors present a systematic empirical study of the effect of planning and model learning on generalization performance, using the MuZero agent. They use 2 environments, Procgen and Meta-World, to respectively explore procedural generalization to new variants of the environment with the same reward structure and task generalisation to new structures of the reward function in the same environment. Their main contributions are their empirical results, specifically that additional reconstruction or self-supervised losses enable the MueZero agent to achieve state-of-the-art performance in procedural generalization, but are not enough to promote a similar increase in performance in task generalization tasks in Meta-World. Finally, they also explore the data diversity dimension, showing that having more diverse data during training can help procedural generalization even more.

**Summary Of The Review:**

The work is very clearly presented, easy to understand and presents a number of ablation cases on some important considerations for RL agents, e.g. model-free, planning and model learning. The learnings from the work are clear, though not necessarily very surprising.

---

> ### Author Response · Authors · 2021-11-12
> **Response to Reviewer hXT2**
>
> Thanks a lot for your review and we’re glad that you found the findings clear and well presented. We agree that some findings in the paper can be considered folk-wisdom, but believe they are important to be experimentally validated and documented. At the same time, a lot of findings were definitely surprising to us, for example:
>
> * We were surprised to see that self-supervision improves generalization even when controlling for training performance: as far as we are aware, this has not been previously documented. Reviewer mZzy also appreciated this finding, writing that: “SSL objectives can help generalization performance on unseen environments, even when they do not improve performance on the training environments, which I found surprising but useful.”
>
>
> * We also did not expect to see such a strong interaction effect of self-supervised losses with data-diversity, with their efficacy becoming more apparent with higher degrees of data-diversity. Some of our earlier experiments also showed that adding reconstruction did not help in single task settings like Atari; it was therefore quite surprising and interesting to us to find that it helped so much in ProcGen!
>
>
> * Recent work has argued that self-supervision should be unnecessary for model-based RL: all that should be required is for models to learn the task dynamics, not necessarily environment dynamics [1]. We observe however that it might not be true for procedural generalization: self-supervised losses provide upto a 4x improvement in data-efficiency compared to MuZero as well as significant improvements in final performance. In light of this prior work, we feel our results are surprising and informative.
>
> [1] Grimm, C., Barreto, A., Singh, S., & Silver, D. (2020). The value equivalence principle for model-based reinforcement learning. NeurIPS 2020.

---

### Official Review · Reviewer_KYiU · 2021-11-02

**Correctness:** 2
**Technical Novelty And Significance:** 2
**Empirical Novelty And Significance:** 2
**Recommendation:** 6
**Confidence:** 4

**Main Review:**

Strengths

- While prior works have shown some sort of self-supervised representation learning to be important in learning world models [1], this work seems to be the first to evaluate its effect on generalization. Since multi-task RL and generalization seems to be an active research area, I think this is a valuable contribution.
- The paper shows that some conclusions as to generalization performance cannot be drawn when using a small training set, which suggests researchers should evaluate their methods on diverse sets of environments to see whether the method affects generalization.

Weaknesses

- The paper talks about procedural generalization and task generalization, where the former corresponds to generalizing to unseen configurations of an environment with the same reward function and the latter corresponds to unseen reward functions in the same environment. Examples of each include ProcGen and ML-1 for procedural generalization and ML-10 and ML-45 for task generalization. To me, it is unclear that ML-10 and ML-45 exactly fit into the category of "task generalization", since there are different types of objects (different environments) in the different task instances.
- ML-10 and ML-45 have significantly fewer training environments than the proc-gen tasks, where the paper claims that the improvement of self-supervised representation learning is less apparent with lower data diversity. Perhaps the results on task-generalization are simply due to this? I am curious as to whether there is truly a difference in generalization performance when measuring generalization to new reward functions rather than new environment configurations. Perhaps there is a better environment to test this in where the data diversity can be controlled for?

[1] Babaeizadeh, Mohammad, et al. Models, Pixels, and Rewards: Evaluating Design Trade-offs in Visual Model-Based Reinforcement Learning. [https://arxiv.org/abs/2012.04603](https://arxiv.org/abs/2012.04603)

**Summary Of The Paper:**

This paper evaluates how well model-based RL, specifically MuZero, generalizes in comparison to model-free RL. It empirically compares how planning, representation learning, and data diversity affect the generalization of agents. To evaluate the effect of planning, a Q-learning agent is constructed to be as similar as possible to MuZero without the MCTS. In experiments, it is found that planning, self-supervised representation learning (reconstruction, contrastive, self-predictive) and data diversity all improve generalization performance. However, results are not similar in the Meta World benchmarks, where self-supervision did not seem to improve results much. The paper concludes that self-supervision is a promising approach to improving the generalization of MBRL agents in procedural environments, but perhaps it does not improve task generalization.

**Summary Of The Review:**

This paper shows that self-supervised representation learning not only improves the training performance of MBRL but also the generalization performance and also points out that researchers may want to evaluate their algorithms on tasks with high data diversity. However, I feel as though the argument that task generalization differs from procedural generalization is not well supported by the experiments. I do not recommend accepting this paper in its current form.

---

> ### Author Response · Authors · 2021-11-12
> **Response to Reviewer KYiU**
>
> Thanks a lot for your review and raising valuable points. We respond to some of your specific comments below. If there is anything else you think we could do to address your concerns and improve the paper, please let us know.
>
> > Do ML-10 and ML-45 exactly fit into the category of "task generalization"?
>
> You are correct that there is some (limited?) procedural variation in the task-generalization setups of ML-10 and ML-45, such as different goal locations or new objects being introduced. However, the environment is coherent with the same setup of the robot arm and the same underlying physics. Out of all existing benchmarks, MetaWorld fits closest to the task-generalization setup we had in mind. It’s also close to what transferring to new tasks means in the real world, we do often come across new environment conditions or objects (for example consider humans learning Tennis on an outdoor court after mastering how to play Squash inside). This does mean however that the boundaries between procedural generalization and task generalization setups are not exact, and we will accordingly modified the MetaWorld section description to say so.
>
> Also note that on ML-45, only 1 out 5 test tasks introduce a new object that the agent hasn’t seen during training. However, 3 out of 5 tasks introduce a new object for ML-10. Thus, we agree that ML-10 might be less useful for testing task generalization. However, we do think ML-45 is still a reasonable choice given that the agent could have explored all the relevant dynamics during training for 4/5 of the test tasks.  We will add some discussion of this in the text.
>
> > Is low-data diversity in ML-10 and ML-45 the reason behind self-supervised losses not showing much improvement there?
>
> This is a great observation, and it could indeed be the case that with large-scale multi-task learning we start to see the benefit of self-supervision for task-generalization again. We looked  for other environments that would facilitate such a large-scale study but we couldn’t find one at the time of writing. In fact, most studies rely on one-off environments to benchmark generalization: MetaWorld is a step better in the sense that it’s well-benchmarked and has multiple pre-training environments. One potential direction could be to let the agents come up with their own tasks during pre-training in an open-ended environment (as done in XLand [1]), which would be an exciting avenue for future work.
>
> > I feel as though the argument that task generalization differs from procedural generalization is not well supported by the experiments.
>
> The purpose in making the distinction between task and procedural generalization was that we intuitively felt there was a qualitative difference between generalization that requires handling new observations versus generalization that requires handling new rewards. We think it is worth studying this distinction and report our results as it is. However, we also agree this distinction is a bit of a fuzzy one, and we don’t mean to make a strong argument that they are fundamentally different from each other (in fact, as described in the first paragraph of 2.1, we define both using the same general meaning of “generalization”). As you point out, the differences we see in performance are confounded by issues such as the amount of data diversity in MetaWorld, so we agree we can’t necessarily draw strong conclusions about whether task generalization and procedural generalization are fundamentally different. We will update the wording in the discussion of our results to reflect that there are multiple conclusions that could be drawn here (e.g., it could be that they’re different and require different approaches, or maybe we just need better environments that expose a larger diversity of reward functions).
>
> [1] Team, O. E. L., Stooke, A., Mahajan, A., Barros, C., Deck, C., Bauer, J., ... & Czarnecki, W. M. (2021). Open-ended learning leads to generally capable agents. arXiv preprint arXiv:2107.12808. https://arxiv.org/abs/2107.12808

---

> > ### Comment · Reviewer_KYiU · 2021-11-22
> > **Thanks**
> >
> > Thank you for the response and the updated paper. I think that the clarifications added as well as changes in writing did a lot to address my concerns about overclaiming a difference between procedural and task generalization. While I think there are probably some experiments that could be done to really evaluate whether there is such a difference, I think this paper does provide value in showing that MBRL excels compared to MFRL in procgen and that MuZero benefits from some form of representation learning (both in training and generalization performance).

---

### Official Review · Reviewer_mZzY · 2021-11-02

**Correctness:** 4
**Technical Novelty And Significance:** 2
**Empirical Novelty And Significance:** 4
**Recommendation:** 8
**Confidence:** 4

**Main Review:**

Pros:
- very clear and well-written paper
- very strong results on ProcGen
- informative ablations and insights
- novel application of model-based methods to procedurally generated environments

Cons:
- limited novelty of methods
- the authors do not mention code release in their reproducibility statement

In section 4.2, is there a reason why MuZero is only trained for 50M while the baselines are trained for 400M steps? It would be helpful to see how MuZero performs when it is run for 400M steps. Or is this too slow? If so, this should be noted somewhere. If it is too slow, then it would be helpful to compare to the baseline performance at 50M steps.

**Summary Of The Paper:**

This paper explores the application of the MuZero agent for tasks which require generalization across environments, namely ProcGen and MetaWorld.
On ProcGen, they find that MuZero in its standard form performs on par or better than strong model-free methods. Furthermore, when combined with auxiliary self-supervised learning (SSL) losses, there is a significant jump in performance which achieves a new state of the art. The paper includes interesting control experiments disentangling the effects of different components. For example, it shows that both MuZero’s modified targets for the value functions, as well as the tree search for action selection, each separately contribute to performance. Another interesting finding is that adding auxiliary SSL objectives can help generalization performance on unseen environments, even when they do not improve performance on the training environments, which I found surprising but useful.

Results are also reported on task generalization benchmarks from MetaWorld. Here the results are less strong, and self-supervision does not appear to help. There appears to be some transfer between tasks, but it is limited.

**Summary Of The Review:**

Overall this is a strong paper and I recommend acceptance. It investigates the application of a model-based RL (MBRL) to procedurally generated environments, which contrasts with most existing MBRL works which on run on singleton environments. In addition to strong results on ProcGen, the ablations are quite informative in understanding the effect of the different algorithmic components. The MetaWorld results are a bit disappointing, but these are nevertheless helpful to include.

My main issue with the paper is that the authors do not mention code release in their reproducibility statement. While the detailed appendix is appreciated, this is not a substitute for releasing code (I am also not aware of thoroughly tested open-source implementations of MuZero). I strongly urge the authors to make their code open source, otherwise it will be difficult for the research community to build on this work.

---

> ### Author Response · Authors · 2021-11-12
> **Response to Reviewer mZzY**
>
> Thanks for your review and valuable feedback!
>
> > Running MuZero for 30M steps instead of 200M steps
>
> You’re right, for computational reasons we decided to run MuZero for 30M steps. The baselines used in prior work build on PPO which is an on-policy algorithm, so their data requirements are much higher than MuZero-Reanalyse which is an off-policy algorithm. The Q-Learning baseline that we report in the paper is the most controlled and appropriate model-free baseline in the paper (same architecture, replay/reanalyse ratios etc.), and we do compare both MuZero and this Q-Learning baseline on the same 30M data budget.
>
> > Would be helpful to compare to the baseline performance at 50M steps.
>
> This is a great idea, we now have access to the raw performance data used in prior works. We will add a plot in the appendix that compares the performance of MuZero and MuZero+SSL methods vs prior work in the 30M data regime.
>
> > Code reproducibility
>
> This is a fair point. We’ve tried to make sure that all new components in this paper (self-supervised losses and their implementation details, new hyperparameters for generalization envs) are adequately detailed. We will also release an implementation of the self-supervised losses themselves in JAX, complementary with a minimal MuZero pseudocode in Python. On the reproducibility of MuZero itself, we note that the EfficientZero [1] implementation should be available soon, and that paper verifies their implementation of  MuZero on established benchmarks like Atari.
>
> [1] Ye, W., Liu, S., Kurutach, T., Abbeel, P., & Gao, Y. (2021). Mastering Atari Games with Limited Data. NeurIPS 2021.

---

> > ### Comment · Reviewer_mZzY · 2021-11-29
> > **Thanks, but still not convinced this is easily reproducible**
> >
> > Thanks for clarifying the first two points.
> >
> > Concerning reproducibility, I'm still not convinced this work will be easily reproducible. I would like to point out that among the 3 third-party MuZero implementations linked to by the authors, the first has only been tested on CartPole, according to the repo (!), the second claims to run on Atari games but does not compare numbers with the original MuZero paper to check that they match, and the third is not yet open source at the time of review. And this is only the base algorithm, not even the experiments in this paper which include SSL losses. It should be the authors' responsibility to make their work easily reproducible, not others.

---

> > > ### Author Response · Authors · 2021-11-29
> > > **Response**
> > >
> > > Our implementation of MuZero relies on internal infrastructure and APIs that are not publicly available, which unfortunately prevents us from releasing the full code. In lieu of this, we have offered to provide as much information and code as we can to facilitate reproducibility.

---

### Official Review · Reviewer_Nzu6 · 2021-11-03

**Correctness:** 4
**Technical Novelty And Significance:** 3
**Empirical Novelty And Significance:** 4
**Recommendation:** 8
**Confidence:** 2

**Main Review:**

The paper systematically investigates how MuZero, a SOTA model-based agent, performs on procedural generalization (in Procgen) and task generalization (in Meta-World). They identify three factors that improve procedural generalization through careful experiments. Next, they test their effect for task generalization and find only a weak positive transfer to unseen tasks. However, the experiments and results testing the effect of planning and data diversity for task generalization were hard to follow.

**Summary Of The Paper:**

This paper measures the generalization ability of model-based agents, i.e. MuZero, in comparison to model-free agents. The authors identify three key factors for procedural generalization: planning, self-supervised representation learning, and procedural data diversity. However, they find that these factors do not necessarily provide the same benefit for task generalization. They argue for a move towards self-supervised model-based agents trained in rich, procedural, multi-task environments.

**Summary Of The Review:**

The authors provide a thorough evaluation of MuZero, a model-based agent, on procedural generalization in Procgen and task generalization in Meta-World. They identify three factors that improve procedural generation and demonstrate challenges in task generalization.

---

> ### Author Response · Authors · 2021-11-12
> **Response to Reviewer Nzu6**
>
> Thanks a lot for your review and feedback, you’re right that the presentation in the task generalization section could be improved. We can reorganize the task generalization section to make the role of planning and data-diversity more explicit. We would love to know if there’s anything more specific we could do to make that section clearer.

---

### Author Response · Authors · 2021-11-19
**Updated version after reviews and feedback**

We thank all the reviewers for their valuable feedback and helpful suggestions. We've uploaded a new version of our paper with the following key changes:
* Reorganized the task generalization section (Section 4.2) to make the role of planning and data-diversity more explicit, as suggested by Reviewer Nzu6.
* Added a plot (Fig A.2 in the appendix) that compares the performance of MuZero and MuZero+SSL methods vs prior work in the 30M data regime, as suggested by Reviewer mZzY.
* Added a note about boundaries between procedural and task generalization not being always exact (Section 2.1), and pointed out the procedural generalization elements in MetaWorld (Section 2.1, footnote), as suggested by Reviewer KYiU.
* Pointed out that the data-diversity in MetaWorld might be too small (especially compared to Procgen) and that studying task-generalization in a high data diversity regime like XLand could be exciting for future work (suggested by Reviewer KYiU)
* Pointed to newer open-source versions of MuZero such as EfficientZero. We also plan to release the code for our self-supervised losses along with a minimal MuZero pseudocode in Python.
* Added a new plot (Fig A.2 in the appendix) comparing the performance of different methods and baselines on newer metrics proposed in [Agarwal et. al (2021)](https://arxiv.org/abs/2108.13264), along with bootstrapped confidence intervals for each method.

---

### Decision · Program_Chairs · 2022-01-20

**Decision:**

Accept (Poster)

**Comment:**

The paper evaluates the generalization capabilities of model-based agents, in particular, MuZero, compared with model-free agents. Reviewers agree that the paper is well-written and the topic is interesting. The ablation study is especially interesting, as it disentangles the effect of different algorithmic components. Some concerns are raised about the significance of this work, as the scope is limited to an empirical study and the results are not necessarily very surprising.

Since the paper presents clear results on an important and relevant topic, I recommend acceptance.